# Quantum States for a Minimum-Length Spacetime

Alessandro Pesci 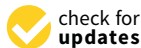

INFN Bologna, Via Irnerio 46, I-40126 Bologna, Italy; pesci@bo.infn.it

**Abstract:** Starting from some results regarding the form of the Ricci scalar at a point $P$ in a (particle-like) spacetime endowed with a minimum distance, we investigate how they might be accommodated, specifically for the case of null separations, in a as-simple-as-possible quantum structure for spacetime at $P$, and we try to accomplish this in terms of potentially operationally defined concepts. In so doing, we provide a possible explicit form for the operator expressing the Ricci scalar as a quantum observable, and give quantum-informational support, thus regardless of or before field equations, to associating with a patch of horizon an entropy proportional to its area.

**Keywords:** quantum gravity; minimum-length metric; small-scale structure of spacetime

## 1. Introduction

From consideration of the combination of gravity and quantum mechanics several results from a variety of approaches have pointed to the existence of a lower-limit length [1–10] (Refs. [11,12] for further references). In recent works an intrinsic discreteness or particle nature for spacetime has been considered by endowing it with a finite lower-limit length $L_0$, built [13–15] in terms of a bitensor also called qmetric effectively embodying this. Among other things, a result has been investigated [14,15] (and Ref. [16] for null separations) consisting in that the Ricci scalar $R_{(q)}$ in the qmetric at a point $P$ does not approach $R$ in the $L_0 \to 0$ limit and has the nature of kind of multivalued quantity, somehow dependent on how we happen to probe spacetime curvature at $P$. The persistence of the effect in the $L_0 \to 0$ limit, is telling that the structure we get in the $L_0 \to 0$ limit (i.e., from a physical point of view, when the effects of a $L_0 \neq 0$ are undetectably small) is not what we have with $L_0 = 0$ (i.e., ordinary spacetime).

Such features, far from being a mathematical accident of the model, are found to unavoidably arise once very basic conditions are met (as the fact that for large separations the qmetric bitensor $q_{ab}$ provides distances which agree with what is foreseen for ordinary metric, and the well-posedness of the Green's function of the D'Alembertian in the qmetric) after input is taken from physics through the requirement of existence of a non-vanishing lower-limit length.

We speculated then that this might be accommodated by assuming that embodying a non-vanishing $L_0$ goes hand in hand with requiring an underlying (finite-dimensional) quantum structure for spacetime at $P$, a consistent description of which would demand on one side a constraint on the metric in the large scale (in the form of field equations) [17], and on another side the existence of some, not better specified up to this stage, quantum operator $\widehat{R}$, corresponding to the observable Ricci scalar, with expectation value $R$ on the maximally mixed state [18].

The aim of present study is to try to investigate further this quantum space that such intrinsically discrete or particle-like spacetime might possess at $P$. This, first of all trying to characterize its states, and then finding an explicit expression for the quantum operator $\widehat{R}$ acting on them. We specifically restrict consideration to the qmetric for null separated events, as in [17,18] on which we elaborate.

Since the multivaluedness of $R_{(q)}$ mentioned above comes about when trying to probe the Ricci scalar reaching $P$ along different geodesics [14–16], a further exploration of this

fact seems to naturally hint to operational notions/procedures. This apparently suggests that a proper framework for further progress might be the consideration of gravity in an as-much-as-possible operational setting, that is in terms of information acquired by physical bodies.

Much progress has taken place in recent times concerning an operational characterization of gravity, in particular regarding the formulation of the equivalence principle, which is at the heart of general relativity, in quantum operational terms, namely with quantum theory considered first of all as a means of information processing. This has been specifically addressed in [19–21], with emphasis in this last work on considering quantum reference frames as something corresponding to real quantum systems (rather than abstract quantum coordinates) as explicitly described in [22]. On top of this, starting from [23,24] several operational procedures have been devised, which might allow in a not far future a direct experimental verification of a quantumness, or non-classicality at least, of the gravitational field, or, quite on the contrary, to rule out any quantum superposition of gravitational fields, along what envisaged in [25].

Also, the strong case has been made that the quantum space which ought to describe gravity is locally finite-dimensional [26], this essentially arising from the Bekenstein bound [27]. On the other hand, finite-dimensional quantum mechanics has been shown to be entirely derivable from a few axioms of general operational theories [28–32].

All this, suggests that in the operational methods there might be all the ingredients needed for a new, deeper-level understanding of the quantum spacetime at a point. This paper is meant as a tentative step in this direction building on the results with qmetric.

## 2. Quantum Spacetime at a Point: Null States

As mentioned, spacetime endowed with a minimum-length $L_0$ exhibits in the small scale a peculiar structure at any event $P$ related to the fact that the qmetric Ricci scalar $R_{(q)}$ does not tend to the ordinary Ricci scalar $R$ in the $L_0 \to 0$ limit. In particular, the limit value depends on the direction of approach to $P$. This is puzzling because the Ricci scalar ought to be determined completely by the assigned spacetime when giving $P$ (with no need of further specifications at $P$); to be sure this ought to be the case at least when $L_0 \to 0$, limit in which all dependence on the direction along which in the qmetric we postulated the existence of an $L_0 \neq 0$ goes to be lost. In this limit one would expect $R_{(q)}$ to not depend on the direction of approach and to definitely be $R$, but as a matter of fact this is not the case. A crucial further feature observed in the Ricci scalar of the qmetric is that if we average the obtained limiting values of $R_{(q)}$ over the possible orthogonal directions we do get $R$.

It appears here sort of similarity between the just described results and what we would get if we went to measure a quantum observable corresponding to the Ricci scalar at (classical) event $P$ with probes reaching $P$ along orthogonal directions, each measurement possibly consisting in taking the expectation value of quantum Ricci according to some suitable state. The purpose of the paper is to try to investigate this similarity and possibly turn it into something more definite and workable on. As we will see below, this is done by introducing an abstract, finite-dimensional Hilbert space $H$ attached to (classical) event $P$, capable of describing all the states of qmetric spacetime at $P$ as well as the observables at $P$, specifically the Ricci scalar.

Having this finite-dim Hilbert space reproducing qmetric results might be enough in that $d$-dimensional Hilbert spaces with assigned finite $d$ are all equivalent to each other. But, which is the physical system which these states are supposed to refer to? It is generically "the spacetime at $P$" as probed by that congruence. But this entails a reference body $A$ (the observer) and a test body $B$ reaching $P$ along the possible geodesics, both generically quantum. We see on one hand minimum length results hint at a kind of microstructure for spacetime, on the other the probing of this possible microstructure means to resort to observers and test bodies. This quite naturally leads to join the minimum-length results with an operational view of gravity.

Following Einstein himself, we consider a point $P$ in spacetime not merely as a mathematical entity in an abstract manifold, but as something physically defined by some crossing or coincidence event among material bodies. In the same spirit, and following the emphasis drawn on this for a long time now (see [33]), we consider as reference frames not merely a set of local coordinates, but actual bodies, quantum bodies since we are going to a small scale, in terms of which the motion of other quantum bodies, the test particles, is described. Point $P$ is regarded as a coincidence event involving a material reference frame (system $A$), generically, but not necessarily, part of the matter which sources the gravitational field, and a test particle (system $B$). In general, we can assume that both the reference body and the test particle give, if any, only a slight perturbation of the field; this, in the perspective of performing measurements disturbing as little as possible. It is clear however that in the vacuum this cannot apply: in this case, the material frame and the test particle are non-negligible sources of the field which is measured at $P$.

In fact, what we would like to do is to explore this way the short scales (in principle, down to the Planck scale), and going to very short scales implies test and/or reference particles of very high momentum, whose own field we can naively expect to definitely dominate at $P$ even in the presence of matter as a source. This would apparently deprive of any possibility of success the attempt to measure in the short scale the properties of a given configuration at $P$.

We have to consider however that, as we will see in more detail below, the departure from the classical result in the expression of Ricci scalar in the qmetric, is at leading order insensitive to the value of $L_0$. The physical effects we want to study in this work are exactly this 'something' which is present due to an $L_0 \neq 0$, and yet stays unchanged from an $L_0$ (relatively) large down to an $L_0$ vanishingly small. To try to investigate which might be the characteristics of this 'something', we assume we can operationally probe it; this is what we are supposing in this work. This amounts to assume that $L_0$, which is a free parameter of the model, is (relatively) large. We have to bear in mind however that with this we are not saying that $L_0$ is necessarily (relatively) large: we are only taking advantage of a not exceedingly small $L_0$ to study some possible consequences of an effect which is scale-independent in the small scales.

How large might $L_0$ be taken? We will see that, concerning the qmetric taken alone, a relevant reference scale is $L_R = 1/\sqrt{R_{ab}l^a l^b}$ (with $R_{ab}$ the Ricci tensor and $l^a$ the tangent to the geodesic under consideration), kind of curvature length scale much larger than the Planck scale. We can consider scales $\ell \ll L_R$ for $L_0$ (but with still $\ell \gg L_{Pl}$) and as test particle e.g., a photon with wavelength $l \sim \ell$, such that the energy density $\rho_\gamma$ of the photon (in a cube of edge $\ell$) is $\rho_\gamma \ll \rho_s$ where $\rho_s$ is the source at $P$ of the field. This is for sure possible if e.g., the source of gravitational field are massive particles.

We emphasize however that the configuration we are considering, even if with an operational flavor, is in the form of a gedankenexperiment. What we are using is a theoretically viable setup to highlight some consequences of the qmetric approach. The actual feasibility of it is another story. As for the latter, the absence of any signal of a $L_0 \neq 0$ at Large Hadron Collider, suggests indeed to take $L_0 < 1/10 \text{ TeV}^{-1} = 2.0 \cdot 10^{-5}$ fm (cf. [34]) (which is in the intermediate scales $\ell$ above). This means that as body $A$ we need mass-energies $m > 10$ TeV and we have to approach $A$ at $\approx$ fm scales (thus $A$ has to be structureless at these energies). To meet these conditions in a controlled way we have to think to colliders, and we can generically expect that the theoretical phenomenon described here might have chances of actual experimental scrutiny as soon as signs of a $L_0 \neq 0$ become visible at colliders.

Summing up, what we do here is –leaving for the moment the actual experimental verification apart– to describe the spacetime at $P$ with the qmetric with $L_0$ in the intermediate range mentioned above. And the physical system to which the Hilbert space above refers (namely the spacetime at $P$) ought to be the composite system made up of $A$ (material reference body or observer) and $B$ (test particle). From basic principles of quantum physics [35] we have then that this Hilbert space $H$ is the tensor product $A \otimes B$,

denoting for simplicity of notation with the same letter the state space and the system to which it refers in each case.

To proceed, we have to specify the states of $A$ and $B$ and of $H = A \otimes B$. To this aim let us first recall in some more detail the results obtained through the qmetric which we would take advantage of in guessing for a quantum description.

As mentioned in the introduction, we here consider the case of null separations. This implies that, in the consideration of the Ricci scalar we are restricting attention to spacetimes (including reference body + test particle, or from these two alone in case of the vacuum) such that a congruence of null geodesics in all spatial directions from $P$ has in it all what is needed to fix the Ricci scalar $R$ at $P$ (in spite of being the congruence short of one dimension as compared to the spacetime $M$) [16]. In these circumstances, using a congruence of null geodesics emerging from $P$ parameterized by length according to a local observer at $P$ which plays the role of reference system $A$, from the relation ($M$ $D$-dimensional)

$$R = \sum_{i=1}^{D-1} R_{ab}\, l^a{}_i\, l^b{}_i, \tag{1}$$

for $L_0$ small we definitely get [18]

$$\frac{1}{D-1} \sum_{i=1}^{D-1} R_{(q),\, l^a{}_i,\, L_0} = R$$

$$= \frac{1}{D-1} \sum_{i=1}^{D-1} R_{(q),\, l^a{}_i}. \tag{2}$$

Here, $R_{ab}$ and $R \equiv g^{ab} R_{ab}$ are the (ordinary-metric) Ricci tensor and Ricci scalar at $P$ (quantities denoted without index $(q)$ refer to ordinary metric); $l^a{}_i$ is the (null) tangent vector at $P$ to the geodesic $i$, with the geodesics taken in spatial directions orthogonal to each other; and [16]

$$R_{(q),\, l^a,\, L_0} = (D-1)\, R_{ab} l^a l^b + \mathcal{O}\!\left(\frac{L_0}{L_R}\, R_{ab} l^a l^b\right) \tag{3}$$

is the qmetric Ricci scalar at $P$ as probed through geodesic with tangent $l^a$ at $P$ with $L_0$ being the minimum length which characterizes the qmetric spacetime and the magnitude of high order terms taken from [17]; $R_{(q),\, l^a} \equiv \lim_{L_0 \to 0} R_{(q),\, l^a,\, L_0}$. We see that, as anticipated above, the leading term does not depend on $L_0$ and the high order terms are smaller by a factor $L_0/L_R$, involving the length scale $L_R$. As mentioned, Equation (3) shows that a same entity, the qmetric Ricci scalar at $P$, happens to take different values depending on through which geodesic we look at it; and this might be interpreted [17,18] as suggesting that the qmetric spacetime at $P$ should be regarded as a superposition of spacetimes, each with its own metric, a characteristic non-classical feature. As repeatedly noticed, this phenomenon arises in a minimum-length $L_0 \neq 0$ description, yet persists virtually unaltered in the $L_0 \to 0$ limit. It clearly stays there when (reduced) Planck's constant $\hbar \to 0$ in case $L_0$ is something different from the Planck length $L_{Pl}$ with $L_0 \neq 0$ when $\hbar \to 0$. But this happens also even if we think of $L_0$ as vanishing with $\hbar$, like e.g., if it is something proportional to Planck length, $L_0 = C\, L_{Pl} = C\, \sqrt{G\hbar/c^3}$ with $C$ a constant (and $G$ and $c$ Newton's constant and speed of light in vacuum), which gives $L_0 \to 0$ when $\hbar \to 0$. This non-classical structure of spacetime is thus something that, when spelled out in particular in quantum terms, is in any case apparently not $\mathcal{O}(\hbar)$, with the meaning that it is not vanishing when assuming a vanishing $\hbar$. It looks like having a status akin to Bell's inequalities. All this seems to resonate with the results [23,24,36], specifically in the description provided in [37].

Relation (2) exhibits the value of the classical Ricci scalar $R$ as an average over $D-1$ qmetric terms. One way to look at this [18], is to take it as suggesting reference to a $(D-1)$-dimensional quantum space (Hilbert space corresponding to $D-1$ perfectly discriminable

states), with Equation (2) expressing the expectation value of the Ricci scalar, considered as a quantum observable, on the maximally mixed state. We try here to bring this perspective a little further.

Let us consider this quantum space as the $(D-1)$-dimensional Hilbert space $H = A \otimes B$ describing the quantum states (of the system consisting of the spacetime at $P$) associated to null directions at $P$. That is, rooted in the just considered similarities, the assumptions in building $H$ are: (i) $H$ is finite dimensional, the dimension being that of the (sub)manifold swept by the congruence of geodesics used to probe spacetime around $P$ (this is $D-1$ where $D$ is the dimension of spacetime, since we use a congruence of null geodesics from $P$); (ii) we take (spatially) orthogonal directions as associated to perfectly distinguishable states, thus to states orthogonal according to the internal product of the Hilbert space.

We then construct $H$ as follows. Denoting $L = \{k^a \text{ at } P : k^a \text{ null and future directed}\}$, we introduce a correspondence $f : k^a = (k^0, \vec{k}) \longmapsto |\vec{k}\rangle$ from $L$ to an abstract $(D-1)$-dimensional Hilbert space with internal product $\langle \cdot | \cdot \rangle$. For $l^a{}_i = e^a{}_0 + e^a{}_i = (1, \vec{e}_i)$, $i = 1, \ldots, D-1$ (here and hereafter $i, j$ label vectors of the base; $e^a{}_0$ is unit timelike), this gives $f(l^a{}_i) = |\vec{e}_i\rangle \equiv |i\rangle$. We introduce $H$ as the abstract $(D-1)$-dimensional Hilbert space with internal product $\langle \cdot | \cdot \rangle$, obtained as the complex span of the elements $|e_i\rangle = f(l^a{}_i)$, $i = 1, \ldots, D-1$, these being coinceived as pure states defined to be orthonormal according to $\langle \cdot | \cdot \rangle$. This implements the view that, as for the system consisting of the coincidence at $P$ (i.e., of the composition of the reference body $A$ and the test particle $B$) the spatial direction associated to a vector $l^a{}_i$ is exactly definite, and is thus represented by a pure state, i.e., by a vector in $H$.

The bijective correspondence $f : L \to H$ is not a correspondence between vector spaces ($L$ is not a vector space). This ought not to be a problem however, provided no incongruences arise in any linear operation in $L$ which maps $L$ to itself; in this case, what we have to require is that the corresponding operation in $H$ brings to a vector $\vec{k'}$ which is precisely the image through $f$ of the vector $k'^a$ we have got in $L$ as a result of the operation. This however is of course guaranteed by the fact that the vector $\vec{k}$ is part of the vector $k^a$ and thus any linear operation on $k^a$ which brings to a vector $k'^a \in L$ involves a linear operation which brings from $\vec{k}$ to exactly the component $\vec{k'}$ orthogonal to $e^a{}_0$ of the vector $k'^a$. This happens in particular for the case of scalar multiplication by $\eta \geq 0$, and for rotations in the subspace orthogonal to $e^a{}_0$; for both, one easily verifies that the vector one gets in $H$ is the image of the vector we get in $L$, indeed

$$\eta f(k^a) = \eta |\vec{k}\rangle = |\eta \vec{k}\rangle = f(\eta k^a), \tag{4}$$

and

$$Q f(k^a) = Q |\vec{k}\rangle = |Q\vec{k}\rangle = f(\widetilde{Q} k^a) \tag{5}$$

where $\widetilde{Q} = (1, Q)$ with $Q$ a $(D-1) \times (D-1)$ orthogonal matrix expressing a rotation in the subspace orthogonal to $e^a{}_0$.

## 3. A Quantum Observable for the Ricci Scalar

In the previous section, we established a correspondence between null vectors $k^a$ of the local frame of coincidence event $P$ and vectors of the Hilbert space $H$, with every element of $L$ represented in $H$ this way. Our next task is now to be able to describe the Ricci scalar as a quantum observable, namely to express it in the form of a Hermitian operator $\widehat{R}$ of $H$.

The results [17,18], have hinted to that the quantity $R_{(q), l^a, L_0}$ in (3), or its $L_0 \to 0$ limit form $R_{(q), l^a} = (D-1) R_{ab} l^a l^b$, might be taken as the output we get from a probe of the Ricci scalar through a geodesic with tangent $l^a$ at $P$, this in turn tentatively giving kind of a yet to be precisely defined expectation value on the maximally mixed state coinciding with

the ordinary Ricci scalar $R$ at $P$. This exhibits expression (2) as a potential candidate from which to start trying to infer an expression for $\widehat{R}$.

Looking at it, from what we did so far it becomes quite natural to think of simply replacing the null vectors $l^a{}_i$ in the terms

$$R_{(q),\,l^a{}_i} = (D-1)\,R_{ab}\,l^a{}_i\,l^b{}_i,\tag{6}$$

in (2), with their quantum counterparts in $H$, leaving $R_{ab}$ as it is. This corresponds to the presumption that the quantum nature ascribed to the (quantum) Ricci scalar might be captured in the simplest manner by resorting to the null quantum states as replacing the null fields, as well as by the occurrence of the factor $(D-1)$.

Inspecting however the terms $R_{ab}\,l^a{}_i\,l^b{}_i$ or, more generally, quantities of the kind $R_{ab}\,l^a{}_i\,l^b{}_j$, we see that even in this as-simple-as-possible prescription we have to face the problem of how to express the sums over indices $a$ and $b$ in terms of vectors of $H$. Essentially the problem is that of being able to manage the time component $l^0{}_i$ of $l^a{}_i$, that is terms of the kind $R_{00}l^0{}_i l^0{}_j$ or $R_{0\alpha}l^0{}_i l^\alpha{}_j$ ($\alpha = 1,\ldots,D-1$, and in the expression we have implicit sum on the repeated index $\alpha$).

Since the vector $l^a{}_i = (1,\vec{e}_i)$ gets mapped into the state vector $|i\rangle$, we do this by introducing the symbol

$$|l^a{}_i\rangle \equiv \big(|i\rangle,0,0,\ldots,|i\rangle,0,\ldots,0\big) = l^a{}_i\,|i\rangle\tag{7}$$

(no sum on repeated $i$ implied). It denotes a string of $D$ (dimension of spacetime) vectors of the Hilbert space. Index $a$ selects the place in the string ($a = 0,1,\ldots,D-1$); the first entry in the string, the time component in index $a$, has the state vector defined by the remaining entries (the space components in index $a$) (in (7) then the same state vector $|i\rangle$ appears both in place $a=0$ and $a=i$). In general, for $l^a = (1,\hat{k})$ null ($\hat{k}$ versor), we have

$$|l^a\rangle \equiv \big(|\vec{k}\rangle, k_1|1\rangle, k_2|2\rangle, \ldots, k_{D-1}|D-1\rangle\big).\tag{8}$$

Using this, we can write the quantum observable representing the Ricci scalar as

$$\begin{aligned}
\widehat{R} &\equiv (D-1)\sum_{i,j=1}^{D-1} R_{ab}|l^a{}_i\rangle\langle l^b{}_j| \\
&= (D-1)\sum_{i,j=1}^{D-1} R_{ab}\,l^a{}_i\,l^b{}_j\,|i\rangle\langle j|.
\end{aligned}\tag{9}$$

Notice that the operator $\widehat{R}$ is real symmetric, then Hermitian.

The cases of Ricci-flat and of Einstein spacetimes are in view of this result somehow special or peculiar, in that the curvature operator $\widehat{R}$ we get from (9) is for them identically 0. As a matter of fact the qmetric Ricci scalar turns out to coincide with the ordinary Ricci scalar for Ricci-flat spacetimes (and also for Einstein spacetimes in the case of the qmetric based on null separations). We notice however that, while this surely deserves further understanding, it seems to have little effect in a context in which the spacetime is probed operationally with a reference body (and a test particle). The stress-energy tensor of the reference body itself generically guarantees indeed $R_{ab} \neq 0$ and $R_{ab}l^a l^b \neq 0$ at P (both in case it is the only matter present and apparently also if it is part of the source distribution (an exception being the cosmological fluid, namely interpreting the cosmological constant effects as due to a fluid with $p = -\rho$)).

Calculating the expectation value of $\widehat{R}$ over the maximally mixed state $\chi = \frac{1}{D-1} \sum_{i=1}^{D-1} |i\rangle\langle i|$ of $H$, we get

$$
\begin{aligned}
\langle \widehat{R} \rangle_\chi &= \text{tr}(\emptyset \widehat{R}) \\
&= \sum_{i=1}^{D-1} \frac{1}{D-1} \text{tr}(|i\rangle\langle i|\widehat{R}) \\
&= \sum_{i=1}^{D-1} \frac{1}{D-1} \langle i|\widehat{R}|i\rangle \\
&= \sum_{i=1}^{D-1} R_{ab} \, l^a{}_i \, l^b{}_i \\
&= R,
\end{aligned}
\tag{10}
$$

where the second equality is from the linearity of the trace, and the last from Equation (1). Clearly the expectation value is the same whichever is the basis we can have chosen for $H$. We mentioned that the multivaluedness of $R_{(q)}$, with the value depending on the geodesic with which we reach $P$, can be interpreted as suggesting that the spacetime at coincidence $P$ can be interpreted as a superposition of geometries. The value we obtain, namely the ordinary Ricci scalar $R$ at $P$, fits then with what one would expect from randomly probing the quantum Ricci scalar with a flat distribution in direction. The operator $\widehat{R}$ as defined by (9) would be thus a possible explicit expression of a quantum observable corresponding to the Ricci scalar along the lines envisaged in [18].

Since $\widehat{R}$ is Hermitian on a finite dimensional Hilbert space, from the spectral decomposition theorem (see e.g., [38]) we know it is diagonalizable. Being it real, this is accomplished by an orthogonal matrix $Q$. We have

$$
\widehat{R} = \sum_{i'=1}^{D-1} \lambda_{i'} |i'\rangle\langle i'|
\tag{11}
$$

$i' = 1, \ldots, D-1$, with

$$
|i'\rangle = \sum_{i=1}^{D-1} Q_{i'i} |i\rangle,
\tag{12}
$$

and correspondingly

$$
\vec{e}_{i'} = \sum_{i=1}^{D-1} Q_{i'i} \, \vec{e}_i.
\tag{13}
$$

The eigenvalues $\lambda_{i'}$ are given by

$$
\begin{aligned}
\lambda_{i'} &= \langle i'|\widehat{R}|i'\rangle \\
&= (D-1) \sum_{i,j=1}^{D-1} R_{ab} \, l^a{}_i \, l^b{}_j \, \langle i'|i\rangle\langle j|i'\rangle \\
&= (D-1) \sum_{i,j=1}^{D-1} R_{ab} \, l^a{}_i \, l^b{}_j \, \langle i|i'\rangle\langle j|i'\rangle \\
&= (D-1) \sum_{i,j=1}^{D-1} R_{ab} \, l^a{}_i \, l^b{}_j \, Q_{i'i} \, Q_{i'j} \\
&= (D-1) R_{ab} \, l^a{}_{i'} \, l^b{}_{i'}.
\end{aligned}
\tag{14}
$$

Here, the third equality comes from being $\langle i'|i \rangle$ real, which gives $\langle i'|i \rangle = \langle i|i' \rangle$; the last from

$$\sum_{i=1}^{D-1} Q_{i'i} \, l^a{}_i = \left(1, \sum_{i=1}^{D-1} Q_{i'i} \, \vec{e}_i\right) = (1, \vec{e}_{i'}) = l^a{}_{i'}. \tag{15}$$

Equation (14) exhibits the $\lambda_{i'}$'s as the quantities $R_{(q), l^a{}_{i'}}$ we find for geodesics with tangent $l^a{}_{i'}$ at $P$ such that $l^a{}_{i'}$ is $l^a{}_i$ rotated by the matrix $(1, Q)$ with $Q$ that same matrix which describes the rotation from the basis $\{|i\rangle\}$ to $\{|i'\rangle\}$.

Notice that, when diagonalizing $\widehat{R}$, we change the basis in $H$ not the local frame at $P$. What happens in the tangent space at $P$ is that we move from the vectors $l^a{}_i$ to the vectors $l^a{}_{i'}$. Then, each single term $R_{ab} \, l^a{}_i \, l^b{}_i$ goes into $R_{ab} \, l^a{}_{i'} \, l^b{}_{i'}$ and changes in this operation (by contrast with what we would get were our operation a change of local frame: $R_{ab} \, l^a{}_i \, l^b{}_i \mapsto R_{a'b'} \, l^{a'}{}_i \, l^{b'}{}_i = R_{ab} \, l^a{}_i \, l^b{}_i$).

Equation (11) can be read as $\widehat{R} = \sum_{i'=1}^{D-1} \lambda_{i'} P_{i'}$, where the operator $P_{i'}$ is the projector onto the (1-dim) eigenspace of $\widehat{R}$ with eigenvalue $\lambda_{i'}$. These operators are orthogonal to each other and form a projected-value measure of the observable $\widehat{R}$. From basic tenets of quantum mechanics, in a measurement of $\widehat{R}$ immediately after another one which gave as a result $\lambda_{i'}$, we have to find again $\lambda_{i'}$ with certainty.

When reaching $P$ we can generically expect to become maximally uncertain about the direction of approach. Looking at the expression (14) for $\lambda_{i'}$ as glimpsed through the qmetric, things go like if, when reaching $P$, a direction, that corresponding to a specific $|i'\rangle$, is chosen at random. The quantum behaviour would be in that when approaching $P$ we become maximally uncertain about the direction of approach, and in that the measurement of $\widehat{R}$ consists in extracting a random direction among those corresponding to the $(D-1)$ eigenvectors $|i'\rangle$.

Another probe of $\widehat{R}$ at $P$ immediately after, would correspond to a new random extraction of direction. Still, quantum mechanics requires for the new measure that same eigenvalue $\lambda_{i'}$. This in itself poses a difficulty, because, if the extraction is random, we will in general expect $|i''\rangle \neq |i'\rangle$.

At this stage, one possibility might be to hypothesize, somehow by fiat, that the new pick of direction is, for a system already probed, no longer at random. Another one, figured in [17], would be to allow that the pick of direction is still at random (according to what one would basically expect from quantum mechanics), but a specific mechanism, related to that the system has been already probed, would prevent from getting a result different from what already obtained. The mechanism would be in terms of a constraint on $R_{ab}$, and then on the metric, and would involve endowing matter with the capability to affect curvature, in such a way that matter could undo the variation of the curvature one would get when going from $\lambda_{i'}$ to $\lambda_{i''} \neq \lambda_{i'}$. The system would keep this way for the Ricci scalar the value $\lambda_{i'}$ already obtained, and then as a matter of fact would keep staying in the eigenstate $|i'\rangle$. Interestingly, the mechanism takes the form of field equations for the metric $g_{ab}$ thus possibly providing a quantum foundation for them.

Let us take stock for a moment. What we have reached so far is that, building on qmetric results, it seems we can construct a finite-dimensional Hilbert space describing the quantum states of the gravitational field (the metric) at an event $P$, and we have given the explicit form of an operator representing in this space the Ricci scalar. This might be something interesting, in that there are several proposed quantum descriptions of gravity, but they in general do refer to *regions* of spacetime. The peculiarity here is instead to consider gravity in circumstances in which these regions shrink (classically) to a point.

This result might be expected, at least according to some perspectives as e.g., Ref. [26] we mentioned, which do maintain that the number of gravitational degrees of freedom in a local region ought to be finite, and ask for what is the finite-dimensional Hilbert space describing them. In this paper this is attempted in the limit of these local regions shrinking to a point making use of minimum-length results. This can be considered in a sense as a sort of explicit implementation of the ideas of [26] exploiting the fact that with spacetime

endowed with a minimum-length we have something which in the small scale hints to a quantum structure.

## 4. Description in Terms of Component Subsystems: Entropy

In the previous sections we have considered a point $P$ in spacetime as a coincidence event involving two physical systems: a reference body (system $A$) and a (lightlike) test particle (system $B$). To accommodate the results of minimum-length metric regarding the Ricci scalar, we found then appropriate to describe the physics of the spacetime at $P$ in terms of the composite system $A \otimes B$, with states described by the (finite-dimensional) Hilbert space $H$. Our aim here, is to try to gain some description of the states of $H$ in terms of the states of the component subsystems.

As one of the hallmarks of quantum theory, we know that pure states of a composite system can correspond to reduced states for the component systems which turn out to be mixed. Each state $|\psi^{AB}\rangle$ is meant to describe a situation in which the direction at $P$ is exactly given, and the state is accordingly pure. Beside $|\psi^{AB}\rangle$, we consider the states $\rho^A = \text{tr}_B |\psi^{AB}\rangle$ of system $A$ and $\rho^B = \text{tr}_A |\psi^{AB}\rangle$ of $B$ we get when tracing out in each of the cases the other system.

The physical situation we would consider is the following: given circumstances in which the test particle reached $P$ along a definite direction, we would like to describe the record of this in system $A$ regardless of $B$ and vice-versa. The idea is that, even if the direction at $P$ is exactly given for the coincidence, namely for $A \otimes B$, this might no longer be true for $A$ and $B$ taken separately. In other words, and focusing the discussion on $A$ (given its role as reference system), it might not be true that, after the test particle reached $P$, in the reference frame at $P$ its arrival direction is known exactly.

We ask: does the existence of a minimum-length for spacetime–property which, we have seen, might suggest a quantum description for the system of the coincidence at $P$– allow to make definite statements also regarding the states of the reference body $A$? To address this question let us define more precisely what may be meant, from an operational point of view, that the system $A$ finds the lightlike test particle along some direction $\vec{k}$ and thus with tangent to the worldline $l^a = (k, \vec{k})$ ($k \equiv |\vec{k}|$).

Imagine that the measurement of direction is done say from the track left by the test particle in $A$, taken this as some kind of spherical detector, when going to the coincidence limit $P' \to P$, i.e., $\lambda \to 0$, with $P'$ the point at which the test particle is at $\lambda$, which is the affine parameterization of geodesic such that $l^a = dx^a/d\lambda$ ($x^a$ local coordinates) and $P'|_{\lambda=0} = P$. Assume that $A$ is characterized by some angular resolution and that this is captured in terms of some small solid angle $\Omega_{(D-2)}$ along any given spatial direction (the same in any given direction).

According to $A$, geodesics are straight lines in its Lorentz frame. At some assigned $\lambda$, $A$ takes samples in a small solid angle $\Omega_{(D-2)}$ in any direction; corresponding to the direction at which the signal results maximized (maximum number of hits in the small solid angle in that direction) $A$ takes every straight line from $P$ to points in the area $\Omega_{(D-2)}\lambda^{D-2}$ at $\lambda$ as a geodesics possibly describing the track of the test particle; this is the estimate $A$ gives of the actual geodesic of the test particle, based on the hits at $\lambda$ and on angular resolution $\Omega_{(D-2)}$. When shrinking $\Omega_{(D-2)}$ (ideally even approaching 0) it obtains a better estimate, and when doing this with $\lambda \to 0$ $A$ finally gets its measurement of the arrival direction at $P$.

Beside the area $\Omega_{(D-2)}\lambda^{D-2}$, we consider the $((D-2)$-dim) area $a$ transverse to the direction of motion of the test particle at any given $\lambda$ caught by the set of geodesics from $P$ in the small solid angle $\Omega_{(D-2)}$ according to the actual metric (at this stage, still unknown to $A$); these geodesics define $a = a(\lambda)$ along the particle trajectory down to $P$.

What we would like to point out is that whenever it happens that

$$a < \Omega_{(D-2)}\lambda^{D-2}, \tag{16}$$

as we expect for curved spacetime assuming null convergence condition holds, this signals that in a measurement by $A$ characterized by the solid angle $\Omega_{(D-2)}$ there is presence of spurious geodesics, namely of geodesics which $A$ takes as geodesics that trustworthy represent the direction of the test particle within $\Omega_{(D-2)}$ but that actually do not. This can be regarded as the presence of a probability $p \neq 0$ that the geodesics taken by $A$ as representatives within $\Omega_{(D-2)}$ of the true geodesic are in reality not reliable for this.

We can try to define more precisely $p$ as follows. We know the van Vleck determinant $\Delta = \Delta(P', P)$ [39–42] (see also [43–45]) is the ratio of density of geodesics from $P$ at $P'$ between the actual spactime under scrutiny and what would give the flat case [44]. Indeed, from the relation

$$\theta = \frac{D-2}{\lambda} - \frac{d}{d\lambda} \ln \Delta, \tag{17}$$

where $\theta = \nabla_b l^b = \frac{1}{a} \frac{da}{d\lambda}$ is the expansion of the congruence, we get

$$\ln \frac{a}{\lambda^{D-2}} = -\ln \Delta + C \tag{18}$$

with $C$ a constant, which the consideration of flat case identifies as $C = \ln \Omega_{(D-2)}$. This is

$$\frac{a}{\Omega_{(D-2)} \lambda^{D-2}} = \frac{1}{\Delta}. \tag{19}$$

We thus interpret $a < \Omega_{(D-2)} \lambda^{D-2}$ as the presence of a probability

$$
\begin{aligned}
p(\lambda) &\equiv \frac{\Omega_{(D-2)} \lambda^{D-2} - a}{\Omega_{(D-2)} \lambda^{D-2}} \\
&= 1 - \frac{1}{\Delta} > 0
\end{aligned} \tag{20}
$$

for the geodesics within $\Omega_{(D-2)}$ to be mistakenly taken as a guess to the actual geodesic of the test particle. Clearly, all this makes sense as far as there is no caustic along $\gamma$ in the interval we are considering; we can think this is always satisfied provided we take $\lambda$ small enough.

Notice that $p$ turns out to be independent of $\Omega_{(D-2)}$. This implies that if circumstances are such that $p \neq 0$, i.e., we are mistakenly guessing to some extent, this is something which cannot be cured on improving in angular resolution (i.e., on taking a vanishing $\Omega_{(D-2)}$).

To characterize things at $P$, we have to consider the $\lambda \to 0$ limit. From the expansion [43]

$$\Delta^{1/2}(P', P) = 1 + \frac{1}{12} \lambda^2 R_{ab} l^a l^b + \lambda^2 \mathcal{O}\left(\frac{\lambda}{L_R} R_{ab} l^a l^b\right) \tag{21}$$

(with the expression for the magnitude of higher order terms taken from [17]), where $R_{ab} l^a l^b$ is evaluated at $P$ and, we recall, $L_R \equiv 1/\sqrt{R_{ab} l^a l^b}$ is the length scale proper, for the given $l^a$, of the assigned curvature at $P$, for $\lambda$ small we get

$$p = \frac{1}{6} \lambda^2 R_{ab} l^a l^b + \lambda^2 \mathcal{O}\left(\frac{\lambda}{L_R} R_{ab} l^a l^b\right), \tag{22}$$

which, we see, gives $p \to 0$ when $\lambda \to 0$. That is, there is no ineliminable probability to be mistakenly guessing the actual geodesic of the test particle starting from the track.

We can now proceed to inspect what happens to this in a spacetime endowed with a limit length. From the mere fact that such a spacetime foresees the existence of a non-vanishing area orthogonal to the separation in the limit of coincidence $P' \to P$ between the

two points ([46–49] for null geodesics), we can expect that something deeply different is going on in this case.

The minimum-length metric $q_{ab}(P', P)$ with base at $P$ and for $P'$ null separated from $P$ is [48]

$$q_{ab} = \mathcal{A} g_{ab} + \left(\mathcal{A} - \frac{1}{\alpha}\right)(l_a n_b + n_a l_b),$$

where $l^a$ is the tangent to the null geodesic $\gamma$ connecting $P$ and $P'$ and $n^a$ null is $n^a = V^a - \frac{1}{2}l^a$ with $V^a$ the velocity of the observer at $P$ (i.e., of the reference system $A$) parallel transported along the geodesic. All these vectors are meant as considered at $P'$. The quantities $\alpha$ and $\mathcal{A}$ are functions of $\lambda$ given by

$$\alpha = \frac{1}{d\tilde{\lambda}/d\lambda}$$

and

$$\mathcal{A} = \frac{\tilde{\lambda}^2}{\lambda^2} \left(\frac{\Delta}{\tilde{\Delta}}\right)^{\frac{2}{D-2}},$$

where $\tilde{\lambda}$ is the qmetric-affine parameterization of $\gamma$ expressing the distance along $\gamma$ from $P$ as measured by the observer with velocity $V^a$, with $\tilde{\lambda} \to L_0$ in the coincidence limit $P' \to P$. Here $\tilde{\Delta}(P', P) \equiv \Delta(\tilde{P}', P)$, where $\tilde{P}'$ is that point on $\gamma$ (on the same side of $P'$) which has $\lambda(\tilde{P}', P) = \tilde{\lambda}$.

In the qmetric, the expansion $\theta_{(q)}$ is [48,49]

$$
\begin{aligned}
\theta_{(q)} &= \nabla_a^{(q)} l_{(q)}^a \\
&= \alpha\left[\theta + (D-2)\frac{d}{d\lambda}\ln\sqrt{\mathcal{A}}\right] \\
&= \frac{D-2}{\tilde{\lambda}} - \frac{d}{d\tilde{\lambda}}\ln\tilde{\Delta}.
\end{aligned}
\tag{23}
$$

Here, $l_{(q)}^a = (d/d\tilde{\lambda})^a = \alpha \, l^a$ is the tangent to the geodesic at $P'$ according to the qmetric-affine parameterization $\tilde{\lambda}$ and the qmetric covariant derivative has the expression $\nabla_b^{(q)} v_{(q)}^a = \partial_b v_{(q)}^a + (\Gamma^a{}_{bc})_{(q)} v_{(q)}^c$ for any qmetric vector $v_{(q)}^a$, with the qmetric connection given by $(\Gamma^a{}_{bc})_{(q)} = \frac{1}{2}q^{ad}(-\nabla_d q_{bc} + 2\nabla_{(b} q_{c)d}) + \Gamma^a{}_{bc}$ [50] where $q^{ab}$ (from $q^{ac}q_{cb} = \delta_b^a$) is $q^{ab} = \frac{1}{\mathcal{A}} g^{ab} + \left(\frac{1}{\mathcal{A}} - \alpha\right)(l^a n^b + n^a l^b)$.

From $\theta_{(q)} = \frac{1}{a_{(q)}}\frac{da_{(q)}}{d\tilde{\lambda}}$ (where $a_{(q)}$ is the $(D-2)$-dim transverse area according to the qmetric), following the same steps as above with $\tilde{\lambda}$ and $a_{(q)}$ replacing respectively $\lambda$ and $a$, Equation (23) gives

$$\frac{a_{(q)}}{\Omega_{(D-2)}\tilde{\lambda}^{D-2}} = \frac{1}{\tilde{\Delta}},\tag{24}$$

analogous to Equation (19).

Then, we interpret $a_{(q)} < \Omega_{(D-2)}\tilde{\lambda}^{D-2}$ as a probability

$$
\begin{aligned}
p(\lambda) &\equiv \frac{\Omega_{(D-2)}\tilde{\lambda}^{D-2} - a_{(q)}}{\Omega_{(D-2)}\tilde{\lambda}^{D-2}} \\
&= 1 - \frac{1}{\tilde{\Delta}} > 0
\end{aligned}
\tag{25}
$$

to be mislead in taking the geodesics within $\Omega_{(D-2)}$ as a guess to the actual geodesic of the test particle.

Using (22), for $\tilde{\lambda} \ll L_R$ (i.e., we are assuming to be close to the coincidence, and that curvature is not too big and we can have $L_0 \ll L_R$) this gives

$$p(\lambda) = \frac{1}{6} \tilde{\lambda}^2 R_{ab} l^a l^b + \tilde{\lambda}^2 \mathcal{O}\left(\frac{\tilde{\lambda}}{L_R} R_{ab} l^a l^b\right). \qquad (26)$$

At $P$, namely in the $\lambda \to 0$ limit, we see that

$$
\begin{aligned}
p_0 &\equiv \lim_{\lambda \to 0} p(\lambda) \\
&= 1 - \frac{1}{\Delta_0} \\
&= \frac{1}{6} L_0{}^2 R_{ab} l^a l^b + L_0{}^2 \mathcal{O}\left(\frac{L_0}{L_R} R_{ab} l^a l^b\right) \\
&\neq 0,
\end{aligned}
\qquad (27)
$$

where $\Delta_0 \equiv \tilde{\Delta}_{|\tilde{\lambda}=L_0}$. Correspondingly,

$$
\begin{aligned}
a_{(q)} &= (1 - p(\lambda)) \, \Omega_{(D-2)} \, \tilde{\lambda}^{D-2} \\
&= \frac{1}{\tilde{\Delta}} \, \Omega_{(D-2)} \, \tilde{\lambda}^{D-2} \xrightarrow{\lambda \to 0} \frac{1}{\Delta_0} \, \Omega_{(D-2)} \, L_0{}^{D-2} = a_0,
\end{aligned}
\qquad (28)
$$

with $a_0$ the limit $(D-2)$-dim orthogonal area [47–49] in the solid angle $\Omega_{(D-2)}$.

We see, Equation (27) may be interpreted as showing that, in a spacetime with a minimum-length, the reference body $A$ has an ineliminable probability $p_0 \neq 0$ to be mistakenly guessing the actual geodesic at the coincidence $P$. We can understand this as follows. Let us take a direction $\vec{k}$ as exactly known for the coincidence, and thus described by the pure state $|\psi^{AB}\rangle = |\vec{k}\rangle$ of $A \otimes B$ associated to the null tangent $l^a = (k, \vec{k})$. If $A$ takes a measurement at $P$ along this direction $\vec{k}$, the outcome is not deterministic, even were the measurement performed with infinite accuracy. Indeed, at ideal experimental conditions still there is a probability $p_0$ that the system is found in a state $|\alpha'\rangle \neq |\alpha\rangle$, where $|\alpha\rangle$ is the (pure) state of $A$ with direction the assigned $\vec{k}$ and $|\alpha'\rangle$ is the outcome pure state corresponding to the ideal measurement of direction performed by $A$. In other words, the state $\rho^A$ of $A$ corresponding to $|\psi^{AB}\rangle$ would be, prior to the measurement, the mixed state

$$
\begin{aligned}
\rho^A &= \mathrm{tr}_B |\psi^{AB}\rangle \\
&= (1 - p_0) \, |\alpha\rangle\langle\alpha| + p_0 \, \rho'^A,
\end{aligned}
\qquad (29)
$$

where the density matrix $\rho'^A$ on the Hilbert space of $A$ has support in the vector space orthogonal to $|\alpha\rangle$.

Summing up, this result can be expressed as follows. The observer $A$ is supposed to measure the arrival direction of an incoming photon $B$ nominally coming along some (null) direction $l^a$ ($p \to P$ along geodesic with tangent $l^a$ at $P$). Because of the existence of a finite limit area transverse to $l^a$, there is an unavoidable (that is, present even assuming perfect resolution for the measuring apparatus) nonzero probability $p_0$ that $A$ is mistaken in measuring the actual arrival direction of photon $B$ arriving along $l^a$. The observer has thus a probability $(1 - p_0)$ to be correct in guessing the arrival direction and a probability $p_0$ to be not.

On the basis of this we can do a little step more in trying to understand the meaning of a $p_0 \neq 0$. We ask what is the average gain of information $G$ by $A$ in guessing with perfect resolution the arrival direction of the photon nominally arriving along $l^a$. We know it is given by the probability of correct guessing times the info associated to that guessing + the probability of incorrect guessing times the info associated to that other guessing (cf. [38]).

The infos here are log(probabilities) as we require that the info gained in the occurrence of two independent events is the sum of the infos of each event taken alone. We want moreover that the less probable is an outcome, the greater is the gain of information we have in actually getting it. Normalizing to have $G = \ln n$ in case of $n$ different, equiprobable outcomes, we have [38] $G = (1 - p_0) \ln(1/(1 - p_0)) + p_0 \ln(1/p_0)$, which is the Shannon entropy $\mathcal{H}(1 - p_0, p_0)$ (in base $e$) of the two-outcome probability distribution $(1 - p_0, p_0)$.

A meaning for $p_0 = p_0(l^a)$ can be drawn from the first term $I = I(l^a)$, expressing the average gain of info in correct guessing, of the just given expression for $G$,

$$I(l^a) = (1 - p_0) \ln \frac{1}{1 - p_0} \tag{30}$$

(this same term is present also in the expression of von Neumann's entropy $S(\rho^A)$ of state $\rho^A$ of (29) (cf. [38]): $S(\rho^A) = \mathcal{H}(1 - p_0, p_0) + (1 - p_0)S(|\alpha\rangle\langle\alpha|) + p_0 S(\rho'^A) = \mathcal{H}(1 - p_0, p_0) + p_0 S(\rho'^A) = (1 - p_0) \ln(1/(1 - p_0)) + p_0 \ln(1/p_0) + p_0 S(\rho'^A)$, where second equality is from being $|\alpha\rangle\langle\alpha|$ pure). Indeed, in the $p_0 \ll 1$ limit it reduces to

$$\begin{aligned} I(l^a) &= (1 - p_0)\left(p_0 + \mathcal{O}(p_0{}^2)\right) \\ &= p_0 + \mathcal{O}(p_0{}^2) \\ &= \frac{1}{6} L_0{}^2 R_{ab} l^a l^b + L_0{}^2 \mathcal{O}\left(\frac{L_0}{L_R} R_{ab} l^a l^b\right), \end{aligned} \tag{31}$$

where in the last equality we used the explicit expression of $p_0$ from (27).

We see $p_0 \simeq I(l^a)$ for $p_0$ small, and thus $p_0$ has the meaning of average gain of information by $A$ in correct guessing of arrival direction (or correct guessing of state $|\alpha\rangle$ in von Neumann entropy's description). Point is that with perfect resolution the observer $A$ would be supposed to always have correct guessing (thus with no gain in info when finding the nominal value $l^a$), were not for the indeterminacy connected with the existence of finite limit area.

The understanding of $p_0$ as a gain of information and its explicit expression (27) suggests it might be reconnected with horizon entropy. Together with the photon from direction $l^a$ let us consider a local Rindler horizon at event $P$ [51] (then with expansion and shear exactly vanishing at $P$ in addition to an identically vanishing twist from hypersurface-orthogonality) with generator $l^a$ (clearly this horizon is not the null congruence emerging from $P$ we used all along the paper). The variation $\delta a$ of area of a small patch $a$ of horizon at event $P$ can be written as

$$\delta a = \left(\int_{-\bar{\lambda}}^{0} \theta \, d\lambda\right) a = \left(-\int_{-\bar{\lambda}}^{0} \lambda R_{ab} l^a l^b \, d\lambda\right) a = \frac{\bar{\lambda}^2}{2} R_{ab} l^a l^b \, a, \tag{32}$$

where the second equality stems from Raychaudhuri equation $\frac{d\theta}{d\lambda} = -\frac{1}{D-2}\theta^2 - \sigma^2 - R_{ab} l^a l^b$ as applied to the horizon, with the first and the second term in the r.h.s. of higher order with respect to the last. In these expressions, $\bar{\lambda}$ is the width of the small affine interval associated to the crossing of the horizon by the test particle; $R_{ab}$ is the Ricci tensor at $P$. Based on field equations, this area variation is associated with a variation of horizon entropy, which precise expression (Wald entropy [52,53]) depends on the actual gravitational theory under consideration.

But, we see that the expression (32) is very similar to expression (27) for $p_0$, also reported in (31) where $p_0$ finds interpretation as average gain of information of $A$ in correct guessing the arrival direction of the photon. This suggests to reconnect the variation of horizon area (32) with the gain of information by the observer at a single elementary event, this hinting to horizon area possessing an information content prior and regardless of any invoking of gravitational field equations (in the paper we never resort to field equations, thus in particular they do not enter in deriving Equation (31)). In [51] the association between horizon entropy and area is motivated outside gravity by quantum field theory

arguments (entanglement entropy between vacuum fluctuations just inside and just outside the horizon); here it is made on the basis of kind of operational and quantum-information arguments in a limit-length spacetime.

Expression (31) also matches the formula for the number of gravitational degrees of freedom $\ln n_g \propto \left(1 - \frac{L_{Pl}^2}{2\pi} R_{ab} l^a l^b\right)$ given in [54] ($L_{Pl}$ is Planck length and $n_g$ is the density of quantum states of spacetime), used in the statistical derivation of gravitational field equations. This is not surprising since here as there the starting point is the existence of a non-vanishing limit area orthogonal to the geodesic when $P' \to P$ in a spacetime with qmetric, and we borrow from that approach the individuation as key quantity the ratio actual area/anticipated flat space area and its $\lambda \to 0$ limit, first considered in [46]. The difference is in that in [46,54] this has been introduced to capture the number of microscopic dofs of the (quantum) spacetime; here to try to operationally show the mixedness of the quantum state associated to coincidence according to reference body $A$, and the ensuing arising of entropy.

The results presented in this section might be summarized as follows. The quantity $R_{ab} l^a l^b$, ubiquitous in spacetime thermodynamics and responsible for horizon entropy in Einstein's gravity, has been here found to be connected to the presence of unavoidable mixedness ($p_0 \neq 0$) in the states describing spacetime, according to any observer, at the most elementary level of a single (classical) event, thus to kind of intrinsic ineliminable blurring in any observer's vision, due to (the nature of) gravity itself not to limitations on observer's side.

## 5. Conclusions

What we did in the paper, has been to try to explore further the idea that the intriguing results [14,15] (and [16] for null separations) concerning the form of the Ricci scalar in a spacetime endowed with a minimum length somehow might allow to sneak a look at a quantum structure for spacetime at a point. We did so with the conviction that those results, far from being artifacts of a mathematical model, come about as direct expression of a sound (possible) physical request, as it is that of (consistently) requiring the existence of a minimum length.

For this further investigation, we have chosen kind of an operational angle. This being in part motivated by the belief that a most convenient way to proceed in physics is to keep close contact with experiment, through use of (at least in principle) as-well-as-possible operationally defined concepts. It is prompted also by the present flourishing of activity in operational approaches to gravity, with concrete hope of a direct experimental test of a non-classicality of gravity foreseeable in a hopefully not-so-far future (Refs. [23,24] and subsequent proposals).

The results and main conclusions can be summarized as follows. Building on the findings of the qmetric in the coincidence limit $p \to P$ of two events and limit length $L_0 \to 0$, it seems natural and possible to associate to a classical event $P$ a finite-dimensional Hilbert space describing the structure of the qmetric spacetime at $P$. A Hilbert space $H$ at $P$ has been explicitly built as well as an operator version of the Ricci scalar; this Hilbert space describes crossing events at $P$ between an observer (state space $A$) and a test particle (state space $B$) with the latter geodesically approaching $P$ ($p \to P$); $H$ can then be taken $A \otimes B$. It has been found that to pure states of $A \otimes B$ (exactly defined crossings) do correspond (reduced density operator) mixed states of $A$ even with perfect experimental resolution on observer's side, this inherent mixedness arising from the area transverse to the geodesic remaining finite when $p \to P$; related to this, it has been found that there is an ineliminable (that is, present even with perfect angular resolution) probability $p_0$, explicitly computed, for the observer to incorrectly guess the arrival direction. We have seen how this $p_0 \neq 0$ can be reinterpreted as average gain of information of the observer when guessing correctly. Because the expression for $p_0$ is essentially analogous to the area variation of a suitable local Rindler horizon at $P$, we noticed how this might be used to provide an operational motivation for endowing a patch of horizon with an entropy related to the area, and this

regardless of field equations (horizon entropy is not based on use of field equations). In other terms, the qmetric appears to give the means to (operationally) introduce gravitational degrees of freedom before field equations.

We can consider our results in the context of other approaches also aiming to explore the consequences of a finite $L_0$ for spacetime description. In [55] for example (see also [34]) an effective metric is introduced coming as what one gets if in the propagator the source is smeared on a scale length $L_0$. It has been usefully exploited to compute corrections to the metric of black hole solutions due to a limit $L_0$ (getting in particular singularity free spacetimes). At variance with the qmetric, this is an ordinary metric (distance $\to 0$ in the coincidence limit between any two events), even if corrected for effects due to $L_0$. What the qmetric approach adds to this is a handle to investigate the microstructure of the spacetime one gets if the limit length $L_0$ is embodied directly in spacetime (distance $\to L_0$ in the coincidence limit). The study of this microstructure is what we dealt with here.

What we have presented is essentially the observation that a puzzling result in the qmetric might find a description in terms of a quantum structure for spacetime at $P$, provided $P$ is considered operationally as a coincidence event. No emphasis has been put on the direct experimental scrutiny of this phenomenon since as mentioned this seems hard to achieve, at least as long as no signs of a limit-length $L_0 \neq 0$ are found at colliders. We may speculate however that what described might have if true a significance with consequences in a sense also at low-energy lab scales. The described quantum structure brings indeed with it that spacetime at a point $P$ might be considered as a superposition of classical geometries at $P$. The limit length $L_0$ sets the scale at which the quantum features are expected to unavoidably show up, where 'unavoidably' means for spacetime generic. We can expect that for specific spacetimes, the quantum features might show up at much larger scales. For example we can consider the spacetime sourced by a delocalized particle: the point is that if superpositions are unavoidably found at scale $L_0$ for undelocalized sources, it is not unreasonable to expect that they might appear, at larger scales, also due to the delocalization of the source. This spacetime might then consist of a superposition of the classical spacetimes corresponding each to a superposed position of the particle, and the related effects might be in principle detectable also at low-energy lab scales (as in the proposals [23,24]).

**Funding:** This work was supported in part by INFN grant FLaG.

**Data Availability Statement:** All data generated or analysed during this study are included in this published article.

**Conflicts of Interest:** The author declares no conflict of interest.

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
