# Peer review of "Quantum States for a Minimum-Length Spacetime"

_2571-712X, doi:10.3390/particles5040033_

Round 1

Reviewer 1 Report

My opinion is that the paper is not apprpriate for physics journal. It might be more suitable for an information theotrtic jouurnal.

The paper deals with the author's claim that for a metric with inbuilt generalizations to incorporate a minimum length scale, there is an inherent ambiguity in defining an event, as simple as arrival of a particle. My criticisma are:

1) No quantitative estimate has been provided regarding this phenomenon.

2) It is not clear how this phenomenon might manifest in an explicit physics problem, maybe a scattering event, as an example.

3) There is no suggestion/discussion as to how to generalize/extend the established quantum principles to account for this phenomenon.

4) There is a well established mimimum length corrected metric, the Nicoloini-Spallucci metric. The author might compare that with the one studied here.

5) Suggestions about possible experimental observation of this phenomenon would be welcome.

Author Response

              Reply to Reviewer 1
              -------------------

In the following I list the points raised by the referee
with my comments amidst them.
Please notice that any line numbers refer to the old, unrevised version. 
---

|My criticisma are:
|
|1) No quantitative estimate has been provided regarding this phenomenon.

What is described in the paper is understood as a phenomenon
which presents itself at scale L_0.

We can think of L_0 as the Planck length, or something quite significantly
larger. The latter is the assumption we make in the paper 
for the analysis we perform not to be inconsistent at start.

From the non-observation at LHC of any effect related
to a finite L_0, 
an upper limit can be be extracted for it 
of roughly 1/10  TeV^-1.

With this upper limit
the effects described in the paper
can be expected to be experimentally accessible 
in a controlled way only in colliders, 
and any hope to detect them 
is relegated to the time when signs of a non-vanishing L_0 will 
come from colliders in the first place.

The point of the paper however is not in the 
direct detectability of this phenomenon,
but in the investigation of
the possible significance of certain qmetric results
using a setup which,
even if corresponding to an operational approach,
has anyway the character of a gedankenexperiment.

I have tried to make these points more explicit in the paper
(the text added after line 117). 

|2) It is not clear how this phenomenon might manifest in an explicit physics 
|problem, maybe a scattering event, as an example.

The coincidence events we consider in the paper, 
which might in fact be described also as scattering events,
aim only to stress
that spacetime points ought to be considered actually as coincidences,
and should be described in terms of states of bipartite systems
(reference body and test body). 

As a matter of fact, from the present upper limit to L_0,
the possible direct detectability of the described quantum effects
can actually be expected to come only through scattering events
at the highest accessible energies.

This is mentioned in the revised version in text added 
after line 117.

There are however experimental contexts,
not related to collider physics,
in which some (possible, speculative) implications of this phenomenon 
might be relevant even at low-energy lab scales 
(this is made explicit in the comments to point 5 below).

|3) There is no suggestion/discussion as to how to generalize/extend the 
|established quantum principles to account for this phenomenon.

This phenomenon does not appear to require in itself
a generalization or an extension of quantum physics,
but corresponds to recognize (arriving at this from the qmetric side)
that gravity also is quantum 
(thing this that almost all of us think
is true, yet which is still not proven in a lab;
the interesting thing would be that 
the qmetric has something to say at this regard). 

Clearly the fact that gravity, and spacetime with it, is intrinsically quantum 
can be expected to affect e.g. quantum field theory in many ways, but this 
goes beyond the scope of present paper.
In my opinion the presented results are very generic,
at the end basically they amount to say that from the qmetric approach 
one gets hints that gravity at P generic
is quantum and one might also extract
some information on this quantum structure. 
As such, one might try to build on this in many different ways
to go farther.

|4) There is a well established mimimum length corrected metric, 
|the Nicoloini-Spallucci metric. The author might compare that with the one 
|studied here.

The Nicolini-Spallucci metric mentioned by the referee
(if i grasped correctly referee's point) 
i would say amounts to replace the metric tensor associated to a given source
with the metric tensor
one gets if in the propagator the source is smeared  
according to a length L_0
(see Nicolini, Spallucci, Wondrak, 1902.11242;
also Nicolini, 2208.05390). 

In the qmetric the nonlocality at scale L_0
is implemented resorting instead to a 'metric' bitensor
capable of providing a finite distance in the coincidence
limit p -> P between two points p, P.

The end product of the Nicolini-Spallucci approach
is a manifold in which in the coincidence limit 
the distance goes to 0. It is then at any point 
(also at the singularities of the original manifold)
an 'ordinary' manifold,
the effects of the limit length being embodied at an effective level
in the smearing of the sources.

The qmetric bitensor instead diverges in the coincidence limit
at any point (to provide a finite distance in the same limit).
This gives a 'manifold' that when inspected at the smallest scales
is wildly different from ordinary spacetime at any point
even for flat spacetime. 
To acquire some understanding of this
structure we get at the smallest scales is precisely
the aim of this paper.

These (Nicolini-Spallucci and qmetric)
are two different approaches to investigate the effects
of a limit length.

Nicolini-Spallucci i think is well suited to extract 
phenomenological consequences of a finite L_0 
on scales large as compared to L_0 (e.g. corrections
to field equations).  

The qmetric, which embodies the limit length directly in spacetime, 
is instead a tool in principle useful to gain insight
into the microstructure one should expect for a spacetime
which has a minimum length built in.
Note that in the qmetric no quantum nature for gravity 
is assumed at start: what we require is that 
the ordinary spacetime we know has actually in it a limit
length in the small scale.

I thank the reviewer for having pointed out the need
of an explicit comparison with the Nicolini-Spallucci metric,
and I have tried to insert some of the considerations above
in the revised version (1st paragraph of the text added after line 487
(last line of the body of the paper),
also added the two references above in the bibliography).

|5) Suggestions about possible experimental observation of this phenomenon 
|would be welcome.

I think i have commented on this
in the reply to points 1 and 2 above.

What i would add here is a (speculative) comment on the fact that 
in a sense this phenomenon if true
may have consequences also at low-energy lab scales.

What is presented in the paper is essentially the observation that 
a puzzling result in the qmetric (namely the fact that R_(q) at a point P 
does not approach R when the limit length L-> 0) 
might find a description in terms of a quantum structure for spacetime
at P, when P is considered operationally as a coincidence event.

This quantum structure brings with it 
that spacetime at a point P might be considered as a superposition
of classical spacetimes at P.

This would be supposed to be kind of theoretical result 
one gets when trying to accommodate
the qmetric results.

The limit length L_0 sets the scale at which the described 
quantum features are expected to unavoidably show up,
where `unavoidably' means for spacetime generic.

For specific spacetimes,
the quantum features of spacetime might show up at much larger scales.

For example we can consider the spacetime sourced by a delocalized particle.
The point is then that the superpositions one
unavoidably finds according to the paper at scale L_0
would make natural to expect
this spacetime to be made of a superposition of the classical spacetimes
corresponding each to one of the superposed positions of the particle,
and the related effects might be in principle detectable also
at lab scales (as e.g. pointed out in Refs. 11, 12).

These considerations 
have been inserted in the revised version
(in the 2nd paragraph of the text added after line 487). 
-------------------------------------------------------------------------------

Reviewer 2 Report

This work is an attempt to quantize gravity. The author connects the existence of a minimal length with the underlying quantum structure of spacetime and introduces a quantum operator.

In my opinion, there is enough originality for this work to be accepted in this journal. I only have two minor comments:

In line 113, the meaning of $l^a$ should be explained.

In Eq.(1), the meaning of $R$ should also be explained, although it has been mentioned in the Introduction. 

Author Response

              Reply to Reviewer 2
              -------------------

I list referee's remarks 
with my amendments.
---

|In line 113, the meaning of $l^a$ should be explained.

I have inserted at the same line a parenthesis with an explicit explanation
of both R_ab and l^a.

|In Eq.(1), the meaning of $R$ should also be explained, although it has been 
|mentioned in the Introduction.

It is now explained in the revised version right below Eq. (1).

----------------------------------------------------------------------------

Author Response

              Reply to Reviewer 3
              -------------------

The comments are inserted amidst the points raised by the referee.

|In this paper the minimum length L_0 does not appear
|dynamically. This does not make the model attractive. By my opinion, 
|the minimum length should
|appear dynamically, otherwise it remains just another parameter.
|However, this minimum length
|model may be useful as an approximation to the loop quantum gravity.

I agree.

I would only notice that
this might be considered however 
also a strength of the qmetric approach.

The results we get with it
transcend the specific theory of quantum gravity
which foresees a limit length,
and can then in principle be usefully exploited  
to compare and check the different theories 
in their predictions.

The fact of not having a dynamical theory to start with
is of course a strong limitation,
but corresponds also to explore the idea that the problem we have
at stack, namely the combination of gravity and quantum physics
might be so formidable that present theories 
could be inadequate or perhaps 
also at the risk to be at the end in part misleading. 

One result one gets with the qmetric is the possibility
to arrive at Einstein's field equations as a max-entropy
procedure (cf. Ref. 41 of old version), 
without necessarily having to stick to the paradigm
of deriving them as the dynamics of a given Lagrangian
(which at the end might be found 
very far from what would be really needed)
(that is, the philosophy would be to non trust too much
what we already know of dynamical systems
when applying it to gravity,
and test it with what we get
sticking only to some few things we can really trust of spacetime
such as its thermodynamic features and, very likely, the existence
of a limit length).

|1. The author constructs the Hilbert spaces corresponding to different points 
|of the quantum spacetime, eqs. (7)-(9). These Hilbert spaces constructed 
|at different points should be related by some operator that depends on \lambda 
|which is a geodesic parameter that appears in the line 311. 
|Can the author write this operator explicitly?

At the moment I'm not able to write this operator explicitly,
i don't know if it can be written at all in the context of qmetric.

The problem we face can be quickly expressed as follows.

In principle we might think to construct the Hilbert spaces
at points p near P using the qmetric q(p, P) with base P
as given by the equation after Eq. (22).

This allows us to compute tensor and scalar quantities 
at point p which is at \lambda from P 
in the qmetric, in particular for example the Ricci scalar R_q(\lambda).

Problem is that the R_q we get this way at the given p is different
for different choices of the base points P, P', P'', ...
(unless P, P', P'', are so separated from p that
R_q(p, ..) for each of them turns out to be invariably the ordinary
Ricci scalar R).

This difficulty is strongly related to the fact that
the qmetric, by its very construction, is meant to provide
information (coming from existence of a minimum length)
at a point.
That is, the interest and the use
of expressions like the equation after Eq. (22)
is exclusively in taking their $\lambda \to 0$ limit.

Basically, things seem that it is not in the qmetric that we have the tools
to compare the limiting situations in two different points.
We should resort to considerations of different nature
to try to reconnect the (putative) Hilbert spaces at different points.
For example, in the case of a static spacetime from symmetry considerations
we might want to identify (modulo unitary transformations)
the Hilbert spaces corresponding to events with
two different times t and t' at a same spatial location, 
but this does not come from the qmetric,
it can not be, i would say, a *result* of it. 

Even if important, i think,
I find it difficult to insert these considerations/problems in the text;
i don't know how to do it;
it seems to me that the risk is to shift the attention 
of the reader from the basic message of the paper
to considerations at a deeper level on the qmetric itself. 

|2. The Ricci flat spaces and the Einstein spaces (when Ricci tensor is 
|proportional to the metric) should be commented in Eq. (9). 
|Some singularities may appear in these two special cases.

Yes, these cases are indeed
peculiar in that they give, from Eq. (9), a curvature operator
which is vanishing. I'm indebted with the reviewer for stressing
the need to comment them.

As a matter of fact the qmetric Ricci scalar turns out
to coincide with the ordinary Ricci scalar for Ricci-flat spacetimes
(and also for Einstein spacetimes in the case of the qmetric
based on null separations).

We notice however that,
while this deserves further understanding,
it seems to have little effect in a context in which the spacetime
is probed operationally with a reference body (and a test particle).
The stress-energy tensor of the reference body itself
generically guarantees $R_ab \ne 0$ and $R_ab l^a l^b \ne 0$ 
at P
(both in case it is the only matter present and seemingly
also if it is part of the source distribution
(an exception being the cosmological fluid,
interpreting the cosmological constant effects as due to
a fluid with p = -\rho, which is however a very peculiar fluid)).

In a sense, this might be perhaps considered a further support
to the idea to try to understand the results with the qmetric
in an operational way (it provides a further observation angle 
on the results with qmetric).

I the revised version I have tried to insert the comments above
(in the text added after Eq. (9)).

|I think the following papers of V.G. Kadyshevsky dedicated to the theory 
|of the minimum length should cited in the bibliography
|References
|[1] V. G. Kadyshevsky, “On the theory of quantization of space-time,” 
|Zh. Eksp. Teor. Fiz. 41
|(1961) 1885-1894; JETP 14 (1961) 1340-1346.
|[2] V. G. Kadyshevsky, “Fundamental Length Hypothesis and New Concept of 
|Gauge Vector Field,” Nucl. Phys. B 141 (1978), 477-496

In the revised version are now explicitly mentioned
the results i'm aware of 
which gave theoretical support
to the existence of a minimum length (in the original version
I omitted this) including the suggested references above (for which I thank)
(the text changed/added at line 11 (beginning of the paper), and
the new references in the bibliography). 

------------------------------------------------------------------------------

Round 2

Reviewer 1 Report

The paper can be accepted for publication.

Reviewer 3 Report

My comments were taken into account. I cannot say that my question about an operator that may relate two finite-dimensional Hilbert spaces constructed at different points was  answered in a satisfactory manner, however  I recommend this version of the  paper for publication in Particles.